# Heat transport to the Central Arctic is Reduced by the Barents Sea Cooling Machine

Shaun A. Eisner<sup>1</sup>, James A. Carton<sup>1</sup>, Leon Chafik<sup>2</sup>, and Lars H. Smedsrud<sup>3</sup>

Correspondence: Shaun Eisner (seisner1@umd.edu)

Abstract. The Barents Sea is a primary gateway for Atlantic Water entering the Central Arctic Ocean and ubiquitous water-mass transformation on the Barents shelf is key for mitigating increases in heat transport to the central Arctic through the St. Anna Trough. Using a mesoscale-permitting reanalysis spanning 40 years, we derive the first long-term estimate of heat transport through the St. Anna trough, finding that it has increased by 0.15 TW/year since 1980. However, this is only half of the 0.28 TW/year trend in increasing heat transport into the Barents Sea through the Barents Sea Opening. Decomposing the heat transports reveals that these trends are entirely due to warming temperatures at the sections with no discernible trend in the volume transports. We find that a northward migration of the largest heat fluxes from the ocean to the atmosphere have resulted in cooler and denser Northern Barents Shelf Water, mitigating the heat transported through the St. Anna trough. However, despite functioning properly, the "Barents Cooling Machine" has been unable to keep pace with the dramatic warming of the Atlantic Water inflow, resulting in the residual trend in heat transport to the central Arctic. Finally, we present the first observational evidence for the "ocean feedback" hypothesis, indicating that it modulates volume transport into and out of the Barents Sea on sub-decadal timescales.

#### 1 Introduction

The Barents Sea is situated at the entrance to the deep Central Arctic Ocean. Along with the Bering and Fram Straits, this relatively shallow shelf sea is a primary gateway for warm Atlantic Water (AW) flowing northwards. Warm and salty AW enters the Barents Sea through the Barents Sea Opening (BSO), which serves as a major source of heat and salt to the central Arctic (Skagseth et al., 2008) (See Fig. 1). Despite exporting heat to the central Arctic, the Barents has historically served as a "cooling machine", with strong heat fluxes from the ocean to the atmosphere that cool the inflowing AW and transform it into denser, cooler Barents Shelf Water. In recent decades, the Barents Sea has experienced a number of climatic changes which fall under the umbrella of "Atlantification", including loss of sea ice, rapid warming, and "Borealization" of it's shelf ecosystems (Smedsrud et al., 2022; Ingvaldsen et al., 2021; Onarheim et al., 2018; Onarheim and Årthun, 2017). It has been suggested that ongoing warming and sea ice loss in the Barents has partially been amplified by a slowdown of the "Barents Sea Cooling Machine" (Skagseth et al., 2020), while others suggest that the cooling machine may instead be migrating northward (Shu et al., 2021). Any changes in the efficiency of the Barents Sea cooling will have outsized downstream impacts on the

<sup>&</sup>lt;sup>1</sup>Department of Atmospheric and Oceanic Science, University of Maryland, College Park, United States

<sup>&</sup>lt;sup>2</sup>Department of Meteorology and Bolin Centre for Climate Research, Stockholm University, Stockholm, Sweden

<sup>&</sup>lt;sup>3</sup>Geophysical Institute & Bjerknes Centre for Climate Research, University of Bergen, Bergen, Norway

30

25 future warming of the Arctic due to the outflow of Barents Shelf Water into the Arctic Circumpolar Boundary Current (ACBC) which transports Barents Shelf Water and AW from the St. Anna Trough as far as the Amerasian Basin (Rudels et al., 1999; Woodgate et al., 2001; Aksenov et al., 2011). Correctly assessing long-term changes in the Barents Sea cooling system and the effect that these changes have on the properties of Barents Shelf Water outflowing through the St. Anna Trough is therefore necessary to understanding the implications of Atlantification on the broader Arctic.

The Barents Sea is a complex system with decadal-scale air-ice-ocean oscillations (Ikeda, 1990) that requires feedback processes to operate. Smedsrud et al. (2013) evaluated two primary feedback mechanisms working to collectively maintain the "Barents Sea Cooling Machine" - a "Wind Feedback" and an "Ocean Feedback". The wind feedback mechanism postulates that increases in the heat transport of AW entering through the BSO results in greater heat loss from the ocean to the atmosphere, resulting in lower surface pressures and stronger westerly winds which in turn continue to drive increased AW inflow. This mechanism has been adressed using models and observations (Bengtsson et al., 2004), but the increased cycloninc winds were not found to be causing increased AW inflow by Heukamp et al. (2023). The ocean feedback mechanism postulates that the same increases in AW heat transport through the BSO which result in increased heat loss from the ocean to the atmosphere also drive an increase in the formation of dense water. This sets up an SSH gradient between the northern Barents Sea and the Eurasian Basin that drives an increase in transport out of the St. Anna Trough, which by mass continuity must increase the volume transport through the BSO.

Unlike the wind feedback mechanism, there has been little discussion and no direct observational evidence of the proposed ocean feedback mechanism, due primarily to a lack of sustained observations in the Northern Barents in the vicinity of St. Anna Trough. Recent discussions regarding the Barents Sea cooling machine have focused on the warming inflow through the BSO and a reduction in ocean heat loss associated with warmer Barents Shelf Waters (Skagseth et al., 2020). However, there has been speculation that the cooling system has migrated northward, rather than reducing in efficiency (Shu et al., 2021). Again, sparse observations in the Northern Barents limit the ability to assess whether changes to the location or efficiency of the cooling machine manifest in the transport of Barents Shelf Water through the St. Anna trough and ultimately into the central Arctic, a key point we focus on in this study.

Reanalyses present a potential solution to the problem of assessing long-term changes in the export of Barents Shelf Water and assessing the ocean feedback mechanism of Smedsrud et al. (2013). Historically, reanalyses have been used effectively to produce long-term estimates of ocean heat and volume transports (Carton and Santorelli, 2008; Karspeck et al., 2017). Comparison studies however, such as that of Uotila et al. (2019), have shown that many reanalyses lack sufficient vertical and horizontal resolution to adequately capture the structure and dynamics of the polar region. In this study we use the new, mesoscale-permitting SODA4 reanalysis, which builds upon the Regional Arctic Reanalysis (RARE) (Chepurin et al., 2025; Carton and Chepurin, 2023). SODA4 improves upon the vertical and horizontal resolution of existing reanalyses and we assess its consistency with observations in the Barents Sea region. We use SODA4 to study trends in heat transport through the Barents Sea Opening and St. Anna Trough from 1980-2020. Additionally, we assess how changes in the Barents Sea cooling system have affected the heat of outflowing Barents Shelf Water and the evidence supporting the proposed ocean feedback hypothesis.

#### 2 Methods

60

75

#### 2.1 Reanalysis Construction

The SODA4 reanalysis is a global  $\frac{1}{10}^{\circ}$ , 75 vertical level ocean state estimate spanning the period from 1980 to the near present. The base model is a coupled MOM5.1/SIS1 model on an Arakawa B-grid with displaced poles. The reanalysis assimilates temperature and salinity profiles from the 2023 World Ocean Database as well as ICOADS 3.0 and NOAA nighttime L3 Sea Surface Temperatures (Chepurin et al., 2025). Atmospheric forcing is provided by the ERA-5 atmospheric reanalysis (Hersbach et al., 2019). While the reanalysis is available at 5-daily temporal frequency, for this study, the monthly frequency output is used.

Temperature, salinity, velocity, sea surface height (SSH), and mixed layer depth (using the de Boyer Montégut et al. (2004) density criterion) are all obtained directly from the SODA4 monthly output. While total surface heat fluxes are available from the SODA4 reanalysis, in this study they are obtained directly from the ERA-5 reanalysis so that they can be separated into turbulent (Latent & Sensible) and radiative (Short-wave & Long-wave) components.

In order to assess long-term changes in hydrographic properties, heat content, and atmospheric heat fluxes, time means of each quantity are computed over each decade: 1980-1989, 1990-1999, 2000-2009, 2010-2019, and 2020-2024. Heat flux and other rate of change quantities are first annually integrated and the annual integrals are then averaged over the appropriate decadal period, while direct hydrographic properties such as temperature and salinity are simply averaged over the entire decadal period.

In order to assess the coupling in the hypothesized ocean feedback mechanism, the SSH gradient is considered, which supposedly drives the associated geostrophic anomalies in flow through STA. For this assessment, we construct a single numerical quantity which is referred to as the "SSH curl" or as " $\nabla \times$ SSH" in an abuse of notation. We define this quantity as the sum of the zonally and meridionally-directed components of the SSH streamfunction. The "SSH curl" can be interpreted as measuring the strength of the cyclonically-directed geostrophic flow generated by an SSH gradient anomaly.

$$\nabla \times SSH \equiv \frac{\partial SSH}{\partial x} - \frac{\partial SSH}{\partial y} \tag{1}$$

#### 2.2 In Situ Observations

Annual time series of 0-200m depth averaged temperature and salinity are obtained from the Bear Island and Kola section time series (obtained from the ICES database: https://ocean.ices.dk/iroc/) and directly compared to annually and depth averaged temperature and salinity from the SODA4 output (Gonzalez-Pola et al., 2023). Time-mean hydrographic properties of the SODA4 output are also directly compared to the World Ocean Atlas 2023 climatology (Reagan et al., 2023).

## 2.3 Transport & Sections

Net heat and volume transports are evaluated over a number of key sections, including the BSO, the Fram Strait (FS), the St. Anna Trough (STA), and across the ACBC. The extent of these sections are indicated in Table 1 and Figure 1. All transports

90 are depth-integrated to a maximum depth of 1000m. This yields a heat transport  $D_Q$ :

$$D_Q = \int \int_{z=-1000m}^{z=0m} \mathbf{u} \cdot \hat{\mathbf{n}} \rho C_p (T - T_{ref}) dA, \quad T_{ref} = -1.8^{\circ} C$$

$$(2)$$

where  $\mathbf{u}$  is the horizontal velocity vector and  $\hat{\mathbf{n}}$  is the unit normal vector to the surface through which the transport is calculated. In the case of our sections, the unit normal is always either directly zonal or directly meridional, reducing the normal component of the horizontal velocity to either u or v.

While a reference temperature of 0°C has traditionally been used for heat transport calculations in the Barents Sea region (Smedsrud et al., 2010), we adopt a reference temperature of -1.8°C in accordance with Woodgate et al. (2001) and Pnyushkov et al. (2018, 2021) so as to avoid any confusion in transport trends or direction resulting from temperatures in the vicinity of 0°C. -1.8°C is a natural lower limit for the reference temperature, as it is close to the freezing-point. To provide direct comparison with prior estimates regardless of reference temperature, we compute the true heat flux convergence of the Barents Sea, following Smedsrud et al. (2010). The true heat flux convergence is defined as the integrated divergence of heat over the entire volume, but can be rewritten in terms of the sum of heat transports across each of the boundaries using Green's Theorem:

$$\iiint \nabla \cdot \rho C_p(T - T_{ref}) dV = \sum_{i=1}^n D_Q^i$$
(3)

Therefore, the true heat flux convergence to the Barents Sea is independent of reference temperature as long as the volume budget is closed, but the absolute values at each section vary with this choice. In SODA4, the volume budget is approximately closed when considering flow through BSO and the entire northern boundary (19-75°E, 79.5°N), so the true heat convergence in SODA4 is approximately the sum of  $D_Q^{BSO}$  and  $D_Q^{NB}$ , where NB represents the northern boundary. For reference, we also compute the heat convergence and time mean heat transports for the well-known GLORYS12 reanalysis (Lellouche et al., 2021) (See Table A1).

Table 1. Transport Sections

| Section                        | Longitudes | Latitudes |
|--------------------------------|------------|-----------|
| Barents Sea Opening (BSO)      | 19 E       | 70-75 N   |
| St. Anna Trough (STA)          | 65-75 E    | 79.5 N    |
| Fram Strait (FS)               | 3W-10E     | 79 N      |
| Arctic Boundary Current (ACBC) | 90E        | 81.5-83N  |

#### 110 3 Results

#### 3.1 Hydrography

We assess changes in the decadally-averaged vertical profiles of hydrographic properties as well as the change in the 2020s decadally averaged heat content relative to the 1980s. Relative to the 1980s, the 2020s Barents-Kara sea heat content increased by 1.1 ZJ. This is consistent with a steady  $\sim 0.2^{\circ}$ C increase in potential temperature each decade. Notably, the increases in potential temperature appear depth-amplified, particularly in the 2020s, suggesting that increases in heat content are the result of warmer inflows at greather than 50m depths (Fig. 2 a). While overall salinity and potential density decreased by 0.5 psu and 1 kg/m3 respectively (Fig. 2b), the trend is not steady and appears dominated by the signal in the Kara Sea and Northern Barents Sea. In the southern Barents Sea, both salinity and density have increased steadily since the 1980s. Increases in heat content are ubiquitous with the greatest increases in the vicinity of the Barents Sea Opening, St. Anna Trough, and west of Novya Zemlaya (Fig 4a).

To assess the reliability of SODA4, we provide comparison to the annual time series of temperature and salinity at the Bear Island and Kola sections. In the vicinity of the BSO, SODA4 has a cold bias, while in the vicinity of Kola section, it has a slight warm bias. The same pattern of temperature bias is evident when comparing the time mean SODA4 hydrography to the World Ocean Atlas 2023, but is less pronounced in amplitude (see Figure S2). There is a 1-2°C warm bias in the Kara Sea and in the vicinity of Novaya Zemlya, with intermittent regions of 1-1.5°C cold bias throughout the Southern Barents, near the BSO. Despite the evident biases when compared with observations, the annual anomalies of temperature and salinity in SODA4 are well correlated with those present in the Bear Island and Kola section time series (see Figure 3). Additionally, the same upward trends in temperature anomalies that are evident in observations are reproduced in the SODA4 hydrography. It is worth noting that there is a distinct downward trend in salinity anomalies at both Bear Island and Kola section beginning in 2010. This appears consistent with the "great salinity anomaly" identified by Holliday et al. (2020). In SODA4, this anomaly is evident at Bear Island, but is not evident at the Kola section. For this reason, SODA4 deviates from observed salinity anomalies at Kola section by up to 0.2 psu beginning in 2015. This same pattern does not appear in the temperature anomalies.

#### 3.2 Heat Transport

Heat transport through both the BSO and STA have increased over the 40-year period. However, the rate of increase of heat transport through the St. Anna trough (0.15 TW/year) is roughly half that of the increase through the Barents Sea Opening (0.28 TW/year), indicating that at least half of the additional heat advected into the Barents Sea is being lost (Fig. 4b,c). The additional heat that does get transported through STA appears to be the primary source of increased heat in the core of the ACBC, which carries Barents Shelf Water exiting through STA along the Eurasian shelf. There is a clear increasing trend in the average temperature of water transported through the ACBC section (warming by almost 1°C over the entire 40 year period), but no discernible trend in heat transport through Fram Strait (Figure 5). This indicates that the increases in heat transport through the STA are the primary advective source of increased heat transported to the central Arctic.

**Figure 1.** The Barents Sea bathymetry and the analyzed sections; The Barents Opening (BSO), Fram Strait (FS), St. Anna Trough (STA), the Arctic Circumpolar Boundary Current (ACBC) and the Kola Section (KS). Colored red/purple arrows indicate the path of Atlantic Water as it traverses the shelfbreak, enters, and leaves the Barents Sea. Colored blue arrows indicate outflow of Arctic water via the East Greenland Current. Volume transports of the major inflows and outflows are used from Smedsrud et al. (2010) and Pnyushkov et al. (2018).

**Figure 2.** Simulated Barents Sea properties over time. a) Vertical profiles of spatially averaged hydrographic properties over the entire Barents Sea domain for each decade. Potential temperature increases relatively uniformly each decade and is slightly depth intensified. Changes in salinity and density are largely dominated by decreases in the Eastern Kara Sea and Eurasian shelfbreak due to icemelt and increased river freshwater export.

The true heat convergence to the Barents Sea in SODA4 is 77.6 TW. This is nearly identical to the results of Smedsrud et al. (2010), which estimated a true heat convergence for the Barents Sea of ~73 TW. When referenced to a temperature of -1.8°C, 86.8 TW enters through the BSO and 15.9 TW exits through the northern boundary extending from Svalbard to the easternmost edge of STA in SODA4. The vast majority (11.2 TW) of this exiting heat leaves through the STA itself between 65-75°E. The close matching of the true heat convergence indicates that the temperature biases in SODA4 have little impact on the modelled heat convergence in the Barents Sea.

**Figure 3.** Time Series of annual anomalies of Potential Temperature (Left) and Salinity (Right) at Bear Island (Top) and Kola section (Bottom).

Decomposing heat transport through the BSO and STA reveals that the increases in heat transport are entirely the result of changes in heat content, rather than changes in the volume transport. The average potential temperature of the inflow through BSO has increased by 0.05 C/year, with a 0.02 C/year increase in the average temperature of the outflow through STA, despite no significant trends in the volume transport through either opening (Figure S1).

#### 3.3 Heat Fluxes

The reduced increase in heat transport through STA relative to the BSO suggests that there must be a heat loss mechanism to accomdate the reduction. We assess the change in decadally-averaged turbulent, radiative, and total heat fluxes in the 2020s relative to the 1980s. There is a clear pattern in both the turbulent and total heat fluxes of a reduction in the amount of cooling in the Southern Barents Sea alongside an increase in the amount of cooling in the central/northern Barents Sea (Figs. 6a,c). Therefore, while atmospheric cooling efficiency is decreased in the vicinity of the BSO, there is an overcompensating increase in the northern Barents. This dipolar pattern is largely the result of the wintertime (DJF) heat flux pattern and appears characteristic of ice edge retreat (Fig. S3). When the ice edge retreats, water which was south of the ice edge now experiences warmer surface air temperatures and transfer less heat to the atmosphere, while newly ice-free waters remain warmer than surface air temperatures and are now exposed to intense atmospheric forcing, allowing for increased heat loss to the atmosphere. It is also consistent with the modelled pattern of heat flux changes in Shu et al. (2021).

## 3.4 Dense Water Formation

The ocean feedback mechanism of Smedsrud et al. (2013) suggests that increased heat loss from the ocean to the atmosphere drives increased dense water formation, that then, by way of an SSH gradient, generates an increase in outflow through the

Figure 4. (a) The change in decadally-averaged Ocean Heat Content in the 2020s relative to the 1980s. There are three "hotspots" located at the Barents Sea Opening, just west of Novaya Zemlya, and in the St. Anna trough. The mean ice edge for each decade is indicated. (b) Time series of the "true" heat convergence  $\nabla_h \cdot Q$  of the Barents Sea in SODA4. The heat convergence of Smedsrud et al. (2010) is also indicated. (c) Time series of the temperature and volume decomposed heat transports ( $\overline{T}v'$  and ' $\overline{v}$  through St. Anna Trough (STA) and Barents Sea Opening (BSO). Trend lines are indicated as well. There is no discernible trend in the volume component of heat transport through either opening, but trends in the temperature components of both the BSO and STA heat transports.

St. Anna trough. Skagseth et al. (2020) found that heat loss in the southern Barents had decreased, while the overall density had remained constant. We investigate the change in decadally-averaged density in the 2020s relative to the 1980s. Our results agree with the findings of Skagseth et al. (2020) that density in the Southern Barents has remained relatively constant over the previous 40 years. However, a hotspot of increasing density is evident in the Northern Barents in the vicinity of the new ice

**Figure 5.** Time series of heat transport through the Barents Sea Opening (BSO), St. Anna Trough and Fram Strait. Average temperature along a section of the (bottom) Arctic Circumpolar Boundary Current (ACBC) is also indicated. Linear trends are indicated with dashed lines. There are increasing trends in heat transport through St. Anna Trough and the average temperature through the ACBC section but no discernible trend in heat transport through Fram Strait.

edge (Fig. 7a). This density change is collocated with a depression in SSH which establishes a strong SSH gradient between the northern Barents Sea and the Eurasian Basin (Fig. 7b). The increased density is consistent with an increase in mixed layer depth and decrease in base of ML stratification in the Northern Barents, indicating that the increase in cooling and dense water formation results in increased vertical mixing and erosion of near-surface stratification (Figs. 7c,d).

**Figure 6.** Change in decadally-averaged Turbulent, Radiative, and total heat fluxes in the 2020s relative to the 1980s. Changes in total heat fluxes resemble changes in turbulent heat flux, with increased heat leaving the ocean (resulting in increased cooling) in the northern Barents sea, northwest of Novaya Zemlya, and decreased heat leaving the ocean in the southern Barents Sea (resulting in decreased cooling).

#### 3.5 The Ocean Feedback Mechanism

We assess the correlations between the SSH curl and the outflow through St. Anna Trough over the 40-year time period. Despite little to no trend in the volume transport through STA and a decreasing trend in the SSH curl in the vicinity of STA, there is a negative correlation between the two, with r = -0.87 (Fig. 8a). Additionally, there is a positive correlation between the outflow through STA and the inflow through BSO (r = 0.84) (fig. 8b). To assess the time scales over which this relationship acts, the correlation and associated significance between STA and BSO transport and SSH curl and STA transport are assessed as a function of low-frequency cutoff (Fig. 9). STA-BSO transport correlations fall off sharply after 1-2 years, becoming insignificant beyond the 3.5 year cutoff. Meanwhile, SSH curl-STA transport correlations fall only slightly at 1-2 year cutoffs before steadily increasing to a maximum at a cutoff of 15 years. However, due to the reduced degrees of freedom, these correlations become insignificant after the 8 year cutoff. This indicates that the relationships between STA outflow and SSH curl in the NE Barents remain consistent across most sub-decadal timescales and that the STA outflow - BSO inflow response only occurs at very short (1-3 year), interannual timescales. At timescales longer than 8 years, it is possible that SSH curl and STA transport do have meaningful correlations but the time series is not long enough for such correlations to be statistically significant. Overall, this indicates that the feedback mechanism is not coherent at greater than decadal time scales (fig. 9). The lack of correlation between the BSO inflow and STA outflow at greater-than-interannual timescales is consistent with the findings that there is no significant trend in the volume transport over the 40-year period.

## 190 4 Discussion & Conclusions

The Barents Sea is a key moderator of heat and salt transport to the Central Arctic Ocean. Historically, the Barents has served as a "cooling machine" where incoming warm and saline AW is cooled into denser Barents Shelf Water before emptying into the

200

205

Figure 7. Change in decadally-averaged depth-averaged density (top-left), mixed layer depth (top-right), curl of ssh (bottom-left), and stratification at the base of the mixed layer (bottom-right) for the 2020s relative to the 1980s. SSH anomaly contours are shown in the left panels. A region of increased density and SSH depression is evident in the Northern Barents Sea, northwest of Novaya Zemlya. A region of anomalously low SSH curl is evident in the St. Anna trough. Mixed layer depth and stratification both decrease in the north-central Barents Sea in the vicinity of ice-edge withdrawal.

Eurasian Basin. However, recent studies have questioned the ongoing efficiency of the cooling machine. Despite its importance to the heat budget of the broader Arctic Ocean and understanding the mechanisms driving the Barents cooling machine, there remain no long-term observational records of heat transport exiting the Barents Sea. To address this lack of observational information, we use the SODA4 reanalysis to assess multi-decadal trends in transport and hydrography in the Barents Sea. We also provide the first observational evidence to support the "Ocean Feedback Mechanism", which has been proposed to constitute part of the broader cooling system (Smedsrud et al., 2013).

SODA4 depicts a clear increasing trend in the heat transport through both the Barents Sea Opening (0.28 TW/year) and the St. Anna Trough (0.15 TW/year), which are entirely the result of increases in the average temperature, with little to no change in volume transport over the 40 year period. Of particular note is that despite a decrease in heat loss to the atmosphere in the southern Barents, temperature changes in the 20s relative to the 80s are bottom amplified. This implies that the inflowing water from the Nordic Seas is the primary warming source. The temperature anomalies of SODA4 are consistent with those of the Bear Island and Kola section time series, suggesting that the increase in heat transport as a result of increasing water temperature is also consistent with observations. This trend has contributed to a significant warming of the water in the ACBC

Figure 8. Scatter plots of  $\nabla \times SSH$  in the NE Barents Sea vs. Volume Transport through STA (left) and Volume Transport through STA vs. Volume transport through BSO (right) with no low frequency cutoff. There is a clear linear relationship between SSH curl in the NE Barents and the outflow through STA, as well a linear relationship between the BSO inflow and STA outflow.

Figure 9. Pearson's  $r^2$  and associated significance (p-value) between the low frequency components of the BSO-STA outflows and the  $\nabla \times SSH$ -STA outflow as a function of the low frequency cutoff. A p=0.05 threshold is also indicated. There are maxima for both values at the shortest timescales. BSO-STA correlations fall off sharply after 1 year, becoming insignificant by the 3-year cutoff, while SSH-STA correlations remain high at longer timescales but become insignificant at timescales greater than 8 years.

core, predominantly by reducing the capability of the Barents Shelf Water to cool overlying AW transported from Fram Strait (which has remained fairly constant in its heat transport). Therefore, despite a mitigation in heat exiting the Barents Sea through the STA, the observed trend in STA heat transport still manifests in downstream increases in heat content, with significant implications for the warming of the central Arctic.

The ocean feedback mechanism proposed by Smedsrud et al. (2013) suggested that an increase in dense water formation in the Barents Sea sets up an SSH gradient that drives increased volume transport through the STA and increased inflow

**Figure 10.** Diagram of the observed ocean feedback loop consistent with the results of Smedsrud et al. (2013). The feedback is driven by dynamical processes that occur at sub-decadal timescales.

through the BSO by mass continuity. Our results support this hypothesis but only on sub-decadal timescales (see Fig. 10). At longer timescales, the outflow through the STA is decoupled from both the BSO inflow and the SSH gradient. However, the increase in heat loss to the atmosphere, and dense water formation in the Northern Barents Sea since the 1980s do suggest that the Barents Sea cooling system (including the ocean feedback mechanism) has migrated northward. This has resulted in less efficient cooling and less dense water formation in the southern Barents Sea. This is consistent with the findings of both Skagseth et al. (2020) and Shu et al. (2021) but suggests that there is more to the overall picture. While the poleward migration of the Barents cooling machine has mitigated the increases in heat transport to the central Arctic by almost half, it has not been efficient enough to dissipate all of the increased heat entering through the Barents Sea Opening. We suspect that the proposed Ocean-Feedback mechanism does play a role in dissipating the excess heat, but that it does so through dynamical processes that occur largely at sub-decadal timescales.

These results have significant implications for future understandings of heat transport into the central Arctic. Our results suggest that continued warming of inflowing AW will continue to be at least partially offset by the Barents Cooling Machine but that at some point, the excess warming will become great enough so as to overwhelm the capacity of the cooling machine. Therefore, continued modelling efforts from models capable of adequately resolving the Arctic mesoscale as well as observations in the Barents Sea and Eurasian Arctic are critical for determining the ultimate fate of the cooling machine.

Data availability. Time series from Kola and Bear Island Sections (S and T) are available at https://ocean.ices.dk/iroc/. SODA4 output is available at https://dsrs.atmos.umd.edu/DATA. GLORYS12 was obtained from the E.U. Copernicus Marine Service (https://doi.org/10.48670/moi-327 00021).

Author contributions. SAE: Conceptualization, Methodology, Software, Formal Analysis, Investigation, Data Curation, Writing - Original Draft, Writing - Review & Editing, Visualization. JAC: Conceptualization, Methodology, Data Curation, Writing - Review & Editing, Resources, Supervision, Project Administration, Funding Acquisition. LC: Data Curation, Methodology, Writing - Review & Editing, Validation, Resources, Supervision, Funding Acquisition. LHS: Methodology and Heat Transport calculations, Writing - Review & Editing.

Competing interests. The authors declare there are no competing interests.

- Acknowledgements. We are grateful to the Physical Oceanography Program of the National Science Foundation (OCE1948952) for providing financial support for this work.
  - L. Chafik was supported by the ECO2NORSE project funded by the Swedish National Space Agency (Dnr 2022-00172).

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
