# Peer review of "Heat transport to the Central Arctic is Reduced by the Barents Sea Cooling Machine"

_EGUsphere, 2025_

## Referee Comment (RC1)

Review of the manuscript entitled "Heat transport to the central Arctic is reduced by the Barents Sea cooling machine" by S.A. Eisner, J.A. Carton, L. Chafik and L. H. Smedsrud.

Using a mesoscale permitting reanalysis spanning 40 years, the authors estimate the transports at the Barents Sea opening, the export through the St Anna Trough and the evolution of the temperature in the Arctic Boundary Current. They suggest that the positive trend of the temperature in the Arctic Boundary Current is linked to the trend in the export through St Anna Trough. Furthermore, they also tested the ocean feedback mechanism proposed by Smedrud (2013). The authors present interesting results; however, the methodology needs to be more clarified. Therefore, I recommend major revisions before publication

**Major comments :**

The definition of the "SSH curl" index is not convincing. Why do the authors not use a simple SSH difference instead? The various links in the ocean feedback mechanism should be described in more detail. The authors could compute the time evolution of the volume of dense water in the formation area, or the water mass transformation, to more clearly demonstrate an increase in dense water formation.

The authors should describe how they compute the temperature index of the boundary current. What are the horizontal and vertical extents of the section? Does the section encompass both the Barents Sea Branch and the Fram Strait Branch? The authors could also map the temperature trend at each point or within sub-regions of the vertical section to determine which layer is most affected by the trend.

The list of references is short and should be completed. Here are some examples. For the observation in Saint Anna Trough, the authors should at least cite Schauer et al 2002 (DSR). For the discussion of the origin of the trend in the heat transport at BSO (Wang et al. GRL, 2019). Cai et al. 2022 ERL, for the role of the surface heat fluxes on the Barents Sea warming. Some papers of Arthun could also be cited. Beszczynska-Möller et al. 2012 for the trend in temperature at Fram Strait. Ruiz-Castillo et al. 2023 (JGR) for the structure of the Arctic Boundary Current North of Severnaya Zemlya. Richards et al. 2022 (JGR) for the trend in the temperature of the Atlantic Water core in the Eurasian basin.

The discussion could be more detailed. In particular, the authors find that the trend in the heat transport at BSO is entirely due to warming temperature. How does this result compare with previous study of the heat transport at BSO?

**Minor Comments:**

l.11:" we present the first observational evidence": sounds a bit odd since the authors are using a model.

L.18: "cooling machine" add a reference.

L.19: "into denser cooler Barents Shelf Water". Add a reference.

L.45: already said l.23-24.

L. 62: "coupled model": Could the authors be more specific: ice-ocean coupled model?

L.63 references for "World Ocean Database, ICOADS 3.0 and NOAA nighttime L3 Sea surface temperatures.

L.63. The authors might specify the number of profiles assimilated in the Barents Sea and St Anna Trough.

L.64 Are the ERA5 surface fluxes imposed? Or is the ERA5 surface air temperature imposed and the turbulent fluxes recomputed based on the SST of the SODA model?

L. 72. Since the authors compute difference between the first and the last period why do they not use 2 periods of the same length (i.e 1980-1989 for the first period and 2015-2024 for the second one).

L.77: STA. not defined.

L.78-81: clarify your definition of "SSH curl" or use a simple SSH difference.

L.80 "1000m" specify what does it mean in term of water mass for the Fram Strait Transport. Does the Fram Strait section encompass the EGC? Could the authors be more specific about the method used to compute the heat transport? Furthermore, the transports through Fram Strait could be compared with observations (Beszczynska-Möller et al. 2012).

L.102: It seems that a time derivative is missing in the formulae.

L. 107. Could the authors indicate the NB section in figure 1.

L. 111: The title of the section is a bit general

L. 116: replace "greather" by greater.

L.115. Without any analysis of the surface heat fluxes and heat transports at the boundaries, it is premature to conclude that the increase in heat content is due to warmer inflows.

L. 120. Fig 4a is cited before figure 3.

L. 122: Are the Kola section data included in the world ocean data?

L. Figure S2. Cited before figure S1.

L. 120-132.I suggest to move this section at the head of the paragraph (as a validation of the model).

L.136: replace fig 4b,c by fig.5

L. 140 . Beszczynska-Möller et al. 2012 suggested that the temperature of the AW core displays a trend in Fram Strait. Is this trend present in the SODA model?  It might be easier to compare equivalent quantities. If the heat transport at St Anna Trough is dominated by the temperature variations, it would be easier to directly compare the variations of temperature at St Anna Trough and in boundary current.

Figure 2. Could the authors add standard deviations ?

L.147 The authors could include a reference to fig. 4b.

L.149: The authors could include a reference to fig. 4c

L. 154: accommodate.

L. 155: A  time series of the surface fluxes would help.

Figure 4: Why do the time series stop in 2020?

Figure 6: Indicate that positive values correspond to a heat loss for the ocean.

Figure 7: a mean state of the mixed layer depth could help better understand the changes in MLD.

---

## Author Comment (AC1)

We kindly thank the reviewer for their very insightful and helpful comments. Below we include line-by-line responses to each comment raised by the reviewer. Where applicable, we indicate how the revised manuscript has been/will be modified to address the comment/proposed change.

**Major comments :**

The definition of the "SSH curl" index is not convincing. Why do the authors not use a simple SSH difference instead? The various links in the ocean feedback mechanism should be described in more detail. The authors could compute the time evolution of the volume of dense water in the formation area, or the water mass transformation, to more clearly demonstrate an increase in dense water formation.

A: This is a good point. To fix this, we have instead switched to using the magnitude of the SSH gradient which we think is more straightforward to interpret. We have also added a short summary of our revised explanation of the ocean feedback hypothesis to improve the discussion of the ocean feedback hypothesis.

The authors should describe how they compute the temperature index of the boundary current. What are the horizontal and vertical extents of the section? Does the section encompass both the Barents Sea Branch and the Fram Strait Branch? The authors could also map the temperature trend at each point or within sub-regions of the vertical section to determine which layer is most affected by the trend.

A: Thank you, however, we are a bit confused by this comment. If referring to Figure 5, we note that it is the mean temperature (averaged over the surface of the section) and we provide the extent in table 1. The extent is from 81.5 to 83N which covers both branches of the boundary current. However, to clarify this, we have made reference to table 1 in the caption of Figure 5.

The list of references is short and should be completed. Here are some examples. For the observation in Saint Anna Trough, the authors should at least cite Schauer et al 2002 (DSR). For the discussion of the origin of the trend in the heat transport at BSO (Wang et al. GRL, 2019). Cai et al. 2022 ERL, for the role of the surface heat fluxes on the Barents Sea warming. Some papers of Arthun could also be cited. Beszczynska-Möller et al. 2012 for the trend in temperature at Fram Strait. RuizCastillo et al. 2023 (JGR) for the structure of the Arctic Boundary Current North of Severnaya Zemlya. Richards et al. 2022 (JGR) for the trend in the temperature of the Atlantic Water core in the Eurasian basin.

A: Thank you for including these references. We have attempted to incorporate them into our discussion where possible.

The discussion could be more detailed. In particular, the authors find that the trend in the heat transport at BSO is entirely due to warming temperature. How does this result compare with previous study of the heat transport at BSO?

A: Thank you, we have lengthened the portion of the discussion which addresses the trend in heat transport by discussing comparison to prior works like Wang et al., 2019; Skagseth et al., 2020; Skagseth et al., 2010.

**Minor Comments:**

l.11:" we present the first observational evidence": sounds a bit odd since the authors are using a model.

A: We have changed the wording to better reflect this. It now reads: 'we present empirical evidence for a revised version of the "ocean feedback" hypothesis'

L.18: "cooling machine" add a reference.

A: We have added appropriate references.

L.19: "into denser cooler Barents Shelf Water". Add a reference.

A: We have added appropriate references.

L.45: already said I.23-24.

A: we have removed the redundant line at L45.

L. 62: "coupled model": Could the authors be more specific: ice-ocean coupled model?

A: Thank you. Yes, we meant ice-ocean coupled. This has been added.

L.63 references for "World Ocean Database, ICOADS 3.0 and NOAA nighttime L3 Sea surface temperatures.

A: Thank you, we have now added these references.

L.63. The authors might specify the number of profiles assimilated in the Barents Sea and St Anna Trough.

A: This is a very good point. We have added a line specifying the approximate number of casts assimilated from WOD.

L.64 Are the ERA5 surface fluxes imposed? Or is the ERA5 surface air temperature imposed and the turbulent fluxes recomputed based on the SST of the SODA model?

A: This is a good point. SODA4 initially imposes the surface fluxes from ERA5 but then

recalculates them to adjust for consistency with SSTs and remove bias in the ERA5 output. In the manuscript, we add a line referring the reader to Chepurin et al., 2025 which describes the process.

L. 72. Since the authors compute difference between the first and the last period why do they not use 2 periods of the same length (i.e 1980-1989 for the first period and 2015-2024 for the second one).

A: This is a good point. We have opted to use only the 2020-2024 period due to the strength of the trend in heat content and heat transport, which produces significant changes in heat content or heat flux, even over a 5 year period (heat transport through BSO would be ~1.2 TW greater in the 2020s than the late 2010s based on the 0.23 TW/year trend). We felt that by not including the second half of the 2010s, the differences more appropriately showcased the magnitude of changes in the Barents Sea in recent years. We do not feel that this hinders the interpretation of our results, as we still show time series of the heat convergence and heat transport over the full period, and the focus of our difference plots is to showcase the spatial pattern of the Ocean Feedback mechanism and correlations between each component, and the correlations themselves are computed over the entire time series as well.

L.77: STA. not defined.

A: This has been fixed.

L.78-81: clarify your definition of "SSH curl" or use a simple SSH difference.

A: We have switched to using the SSH gradient in place of the SSH curl to reduce confusion.

L.80 "1000m" specify what does it mean in term of water mass for the Fram Strait Transport. Does the Fram Strait section encompass the EGC? Could the authors be more specific about the method used to compute the heat transport? Furthermore, the transports through Fram Strait could be compared with observations (Beszczynska-Möller et al. 2012).

A: We have included a line which specified our choice of cutoff at 1000m. With regard to the Fram Strait transport, this could mean that the total transport through Fram Strait is underestimated, but the trend would be unaffected. While we agree with the reviewer that comparison to observations would be helpful, we found that including a direct comparison would be a bit outside the focus of the paper since the discussion of Fram Strait is intended to motivate our discussion with regard to the importance of the trend through STA to the boundary current, rather than being it's main focus. This is further complicated by our need to use a different set of boundaries for the Fram Strait section than is used in observations (since we exclude the EGC) which hinders direct comparison. However, we do add a short discussion, noting a temperature trend through Fram Strait, which is smaller in magnitude but of the same direction as that identified by Beszczynska-Möller et al. 2012, to show that our findings are still somewhat consistent with observations in the region.

L.102: It seems that a time derivative is missing in the formulae.

A: We are somewhat confused as to what the reviewer means. This is simply a rewriting of Green's/Gauss' Theorem, which states that the divergence of a quantity (in this case heat) over a given volume is equal to the flux across all surfaces entering that volume (in this case, it is the sum of the heat transports across all the boundaries D_i). This quantity should be time-independent so long as all surfaces are considered.

L. 107. Could the authors indicate the NB section in figure 1.

A: Thank you, the NB section has been added to Figure 1.

L. 111: The title of the section is a bit general

A: We have renamed to "decadal changes in hydrography".

L. 116: replace "greather" by greater.

A: This has been changed.

L.115. Without any analysis of the surface heat fluxes and heat transports at the boundaries, it is premature to conclude that the increase in heat content is due to warmer inflows.

A: This claim has been removed.

L. 120. Fig 4a is cited before figure 3.

A: The sentence referencing Figure 4a has been moved to fix this.

L. 122: Are the Kola section data included in the world ocean data?

A: Yes, both the Kola section and Bear Island section data are included in the world ocean database.

L. Figure S2. Cited before figure S1.

A: The supp. Figures have been reordered to address this.

L. 120-132.I suggest to move this section at the head of the paragraph (as a validation of the model).

A: We are somewhat confused as to what the reviewer is referring to. Line 120-132 seems to refer to an entire paragraph, which does discuss the validation of the model already.

L.136: replace fig 4b,c by fig.5

A: this change has been made.

L. 140 . Beszczynska-Möller et al. 2012 suggested that the temperature of the AW core displays a trend in Fram Strait. Is this trend present in the SODA model? It might be easier to compare equivalent quantities. If the heat transport at St Anna Trough is dominated by the temperature variations, it would be easier to directly compare the variations of temperature at St Anna Trough and in boundary current.

A: Thank you. As noted above, we have added a brief discussion about the trend in temperature through Fram Strait.

Figure 2. Could the authors add standard deviations ?

A: This is a good point. We have now added standard deviations to this figure.

L.147 The authors could include a reference to fig. 4b.

A: This reference has been added.

L.149: The authors could include a reference to fig. 4c

A: This reference has been added.

L. 154: accommodate.

A: This has been fixed.

L. 155: A time series of the surface fluxes would help.

A: While we agree with the reviewer that a time series of heat fluxes would provide further understanding of the trends in heat flux, we have chosen to omit this to preserve the focus of the paper. A proper discussion of the time series of heat fluxes should involve defining regions within the Barents Sea so that we can see the time series in different locations (for example the southwest vs. the northeast) and see how the site of heat loss migrates northward. We felt that defining these locations and providing information with regards to how we chose them would introduce unnecessary length and complexity to the paper given that the change in heat fluxes are not one of the primary key points (instead the main key points focus on the temperature-driven increase in heat transport, heat transport through STA, and the ocean feedback mechanism).

Figure 4: Why do the time series stop in 2020?

A: Thank you for pointing out this oversight on our part. We have fixed the time series to now show the full record (1980-2024).

Figure 6: Indicate that positive values correspond to a heat loss for the ocean.

A: This has been indicated.

Figure 7: a mean state of the mixed layer depth could help better understand the changes in MLD.

A: While we agree with the reviewer that the time mean MLD would help the reader better understand the changes, we show MLD not to illustrate the actual changes with MLD but instead to further substantiate the idea that there is dense water formation since we see the mixed layer is deepening and stratification at the base is eroding, providing some evidence of dense water formation. However, it may be the case that instead these two panels should be removed if the reviewer thinks they do not contribute significantly to the discussion. For now, we have kept them but can remove them if the reviewer prefers.

---

## Author Comment (AC2)

We would like to thank the reviewer for their very insightful and helpful comments on the manuscript. Below we have done our best to address each comment raised by the reviewer and provide a response to each which is highlighted in blue text. Where applicable, we indicate how the revised manuscript has been/will be modified to address the comment/proposed change.

Specific comments:

Title: I think the title needs to be reworked a bit. As it reads now, it is stating the obvious, that a cooling machine cools the heat transport to the Arctic. However, the main finding is that the cooling machine is not fully keeping up the pace of the warming in the Barents Sea region in the sense that there is a positive trend in the heat transport to the Polar Basin, although this trend is less than the trend in the upstream heat transport into the Barents Sea. This should be better reflected in the title, I think.

A: This is a very good point, and indeed our original title was quite obvious. We have changed the title to "Increased ocean heat transport to the central Arctic despite a well working Barents Sea Cooling Machine". This way the title reflects one of the main findings that the reviewer notes, which is that the heat transport through STA has increased and the Barents cooling machine is not keeping pace.

L1: "Central Arctic Ocean" – is this an official name? The deep part of the Arctic Ocean is often referred to as the Polar Basin. It may also be referred to as the central Arctic Ocean, but not with a capital 'C'. And in L2 you refer to the central Arctic with a lowercase 'c'. Please be consistent.

A: Thank you for this catch. We have made the c in central lowercase.

L2: Here and elsewhere: Sometimes you refer to the Barents Sea, and sometimes only to the 'Barents'. I presume you refer to the same, namely the Barents Sea, in both cases. Please be consistent and spell out 'Barents Sea' everywhere you refer to it.

A: Thank you, we have fixed this.

L4: Here and elsewhere: Please spell 'St. Anna Trough' with a capital 'T'.

A: Thank you. We have fixed this.

L5: I think you should change 'increasing' with 'the': "… trend in the heat transport into the Barents Sea …"

A: This has been fixed.

L7-8: Has the migration of the "center of action" of the net heat loss from ocean to the atmosphere resulted in a cooler and denser Barents Shelf Water? Rather, you have shown that

the temperature and density, in general, have increased. Regarding the density of the Barents Shelf Water, see also my comment to lines 169-171.

A: Thank you for this insight. In our use of 'Barents Shelf Water', we were intending only to distinguish from the Atlantic water that enters at Fram Strait and BSO. So when we say that the Barents Shelf Water has gotten cooler, we are stating that on average the water in the Barents Sea has gotten denser and cooler as a whole since an in-depth discussion of individual water masses is not our primary focus. Here and elsewhere, we have refrained from saying "Barents Shelf Water" and said either "outflowing waters from the Barents Sea", "waters exiting the Barents Sea through St. Anna Trough", or other versions of this where appropriate to avoid confusion.

L14: The Bering Strait is not a primary gateway for Atlantic Water to the Arctic Ocean, so the reference to the Bering Strait does not belong here.

A: This error has been removed.

L16-17: While the Atlantic Water carries a substantial amount of heat and salt through the Barents Sea Opening, the contribution of heat to the Arctic Ocean has been questioned (e.g., Gammelrsød et al., 2009), although data sources in the exit region of the Barents Sea are scarce.

A: This is a good point. What we really intended to say was that the Barents Sea is a key moderator of heat and salt so we have adjusted the phrasing to clarify this.

L25-26: On line 19 you state that Atlantic Water are transformed into Barents Shelf Water within the Barents Sea, while here you state that both BSW and AW leaves the Barents Sea and are carried by the ACBC as far as the Amerasian Basin. There is little evidence that water masses leaving the Barents Sea through the At. Anna Trough can still be classified as Atlantic Water.

A: This is a good point and gets back to our misuse of some water mass names that was pointed out earlier. Here, we have instead refrained from referring to 'Barents Shelf Water' or 'Atlantic Water' and instead note that this water that exits through the St. Anna trough will join the boundary current system and be transported downstream.

L38 (and L164-165): This interpretation of the feedback loop proposed by Smedsrud et al. (2013) is not entirely correct. One main argument in the proposed feedback loop is that increased density in the BSX causes a density gradient between the BSX and the Polar Basin, and therefore also an accelerated outflow through the St. Anna Trough. This, in turn, creates an SSH gradient and an associated barotropic forcing through the Barents Sea (from the BSO to the BSX) that favours stronger throughflow.

A: We are slightly confused regarding this comment. I believe what the reviewer is saying is what we were trying to state in this line: that "This [increase in density] sets up an SSH gradient

between the northern Barents Sea and the Eurasian Basin that drives an increase in transport out of the St. Anna Trough, which by mass continuity must increase the volume transport through the BSO.", since the resulting barotropic flow originates from pressure gradient forces which first come from the SSH gradient and then from the mass continuity. To avoid confusion however, we have adopted the reviewer's wording.

L42: "Northern Barents Sea": The region of the Barents Sea that is the focus area of this study is often referred to as the northeastern Barents Sea, whereas "northern Barents Sea" often refers to the area between the Svalbard and the Franz Josef Land archipelagos. Therefore, I recommend referring to your focus area as "the northeastern Barents Sea".

A: Thank you, we have changed the naming to reflect this.

L57: "Barents Sea cooling system" – I would rather use the term "cooling processes".

A: Thank you, this change has been made.

L65: The last sentence in this paragraph clearly belongs to the paragraph below, because I assume you are not using monthly averaged atmospheric reanalysis as forcing for the ocean model!

A: Thank you, this has been fixed.

L77: The acronym 'STA' is not defined yet (it is defined on line 89), although the St. Anna Trough is first mentioned on line 26 (except for in the abstract) and also on several occasions before the acronym is established.

A: We have corrected this so that STA acronym is defined here.

L117: There is a reference to Fig. 2b, but Fig. 2 does not have any panel b as far as I can see.

A: Thank you for pointing this out. This was a mistake and has been fixed.

L119: Note, that the areas with the greatest increases in heat content are also the deepest parts of the Barents Sea (the Bear Island Trough and the Hopen Trench, the Central Basin, and the St. Anna Trough), with the exception being the Northeast Basin. Normalizing the change in heat content by calculating per meter water depth may provide a somewhat different picture.

A: This is a good point brought up by the reviewer. Indeed, the heat content 'hotspots' may be partially the result of topography. After examining the per depth heat content, we have decided to include it in supplemental materials. We retain the actual heat content in the original paper since we find this to be easier to interpret/understand, especially since we are only trying to show the change in heat content in this figure, rather than claim that that change is independent

of the depth. However, to provide the full story, as the reviewer suggests, we include the per depth heat content in supplemental materials.

L124-126: So, a conclusion from this comparison with observations is that the heat loss in the model is smaller than the observed heat loss. I think this observation deserve a bit more discussion considering that the heat loss in the Barents Sea is the main topic of the paper.

A: Thank you for pointing this out. We have added an additional discussion to this section pointing out the importance of recognizing the bias but noting that the trends and anomalies are still very consistent, which allows us to look at trends in heat transport. Also we note the importance that the time mean heat convergence is consistent with observations from Smedsrud et al., 2010 which shows that SODA4 seems to match the observed heat budget in a time mean sense.

L128-129: Change "upward" and "downward" trends with "positive" and "negative" trends, respectively. For clarity, you should also consider removing the reference to anomalies, and instead report that there are trends in the temperature and salinity (and, hence, also a trend in anomalies). Because a negative trend in anomalies could be interpreted as a trend towards more negative anomalies, but it could also be interpreted as a reduction in the magnitude of the anomalies.

A: Thank you, we have fixed this.

L138: I suppose you mean "There is a clear positive trend in the average temperature". An increasing trend, strictly speaking, means an accelerating trend (i.e., a positive second derivative). Please clarify.

A: Thank you. Yes, this is what we meant.

L149: I suggest changing "heat content" with "temperature", because that is what you have shown when decomposing the heat transport in Fig. 4.

A: This has been fixed.

L153: I suggest replacing "The reduced increase" with "The smaller increase".

A: This has been fixed.

L154: Missing 'o' in "accommodate"

A: This has been fixed.

L169-171: You state that you find an increase in the average water column density in the northeastern Barents Sea from the 1980s to the 2020s, and a collocated reduction in the SSH,

which is shown in Fig. 7 (left panels). I have two main concerns with these findings and your interpretation of them. The first concern is related to the effect of the SSH difference on the circulation. A change in the density difference between two regions, here the northern Barents Sea and the Eurasian Basin, also warrants an associated change in the SSH difference between the two regions. Here, the lower SSH in the northern Barents Sea together with the larger density in the same region may cause a net zero change in the pressure difference between the two regions (a higher water column of lower density vs. a lower water column of higher density). Thus, the conclusion that these changes in SSH will cause a geostrophic circulation anomaly is not necessarily substantiated. Moreover, Figure 7, lower left panel, shows a negative "SSH curl" along the eastern flank of the St. Anna Trough (where the Barents Sea Water exits the Barents Sea for the Polar Basin). This would imply a negative contribution to the flow in the cyclonic direction (if I understand the interpretation of the "SSH curl" on lines 79-80 correct). This would be in agreement with a circulation anomaly with the higher SSH on the right-hand side, as also implied by the SSH contours in the panels on the left-hand side in Fig. 7. But this would imply weaker, not stronger, outflow through the eastern St. Anna Trough. And, as a consequence, I have a bit of a hard time following your interpretation of the results depicted in Fig. 7 as synthesized in Fig. 10. Also, the change in density as shown in Fig. 7, upper left panel, seems very large. A density decrease representing a full unit (1 kg/m3) when averaged over the full water column is substantial. And this leads to my second concern. You state, based on Fig. 7, that the density of the Barents Shelf Water and the dense water formation in the Barents Sea have increased. However, such changes will be disguised due to the fact that several water masses are included in your calculation of total water column density change. In the area where you find the largest density increase, in the Northeast Basin in the northeastern Barents Sea, the water column may consist of dense bottom water formed through brine rejection during ice formation, overlaid by Atlantic Water (Barents Sea/Shelf Water), overlaid by Arctic Water. Any change in the mass distribution between these three water masses will also affect the average density of the water column. Indeed, Lind et al. (2018) reported a decline in the amount of Arctic Water in recent decades in this region, which by itself would increase the water column density because the colder and fresher (and less dense) Arctic Water was replaced by warmer and more saline (and denser) Atlantic Water.

A: This is a very good point, and we think it all comes back to our confusing use of "SSH curl". To fix this, we have decided instead to use magnitude of the SSH gradient, which we think shows things much better. With regard to density, to clarify, the 1 kg/m3 number refers to the density change at the surface, but is much smaller when averaged over depth (as can be seen in Figure 7, it is more like 0.3 kg/m3 when depth averaged). With regard to the dense water formation discussion, we certainly acknowledge that the story is complicated by multiple water masses. Based on density alone, we would agree that one could not necessarily conclude that there is more dense water formation. However, we also show that the turbulent heat loss from the ocean to the atmosphere has increased over the same region where density has increased. While this increased heat loss could be caused by an increase in the volume of warm, saline water masses, it also shows that the heat from those water masses is then being extracted and the water is being cooled. So, even if the background density has increased because of a decrease in the ratio of Arctic water to Atlantic Water, there is still overall more cooling (and thus

dense water formation) in the northeast Barents Sea. However, we certainly acknowledge that we do not know what water masses are actually being cooled and becoming denser. For this reason, we try not to specify certain water masses (hence our decision to remove the term 'Barents Shelf Water').

L176-177: Usually, the timeseries are detrended before the correlation analysis is performed. Thus, the different trends in the two timeseries should not affect the correlation between them. Also here, you state a "decreasing trend", while a presume you mean a "negative trend".

A: The reviewer is correct, and indeed the timeseries are detrended before correlation analysis. This was confusing wording on our part. We have left out the part about the trends.

L195-L198: While you have found some support for an ocean feedback mechanism, the underlying hypothesis and the associated analysis is slightly different from that of Smedsrud et al. (2013). Also, your analysis is based on model results, hence, you have not presented observational evidence for an ocean feedback mechanism, but rather empirical evidence, which was also, to some degree, presented in Smedsrud et al. (2013).

A: This is a good point. We have modified this wording here and elsewhere to clarify that rather than observational evidence, we have found evidence for a modified version of Smedsrud's original proposed mechanism.

L199: Again: "increasing trend" -> "positive trend".

A: This has been fixed.

L210-211: The hypothesis proposed by Smedsrud et al. (2013) postulated that an increase in the dense water formation in the BSX area will cause a density-driven acceleration of the outflow through the St. Anna Trough, and a subsequent increase in the BSO inflow due to mass continuity.

A: As with the previous comment regarding the description of the feedback mechanism, we have adopted the reviewer's wording to avoid confusion.

L212: The reference to Fig. 10 should be a reference to Fig. 9, I believe.

A: This is correct, thank you. This has been fixed.

L215: Again, I would rather use the term "cooling processes".

 A: Thank you. We have changed this.

Figures:

Figure 3: I think you got the references (left/right and top/bottom) to the variables and geographical locations mixed up. Please correct. Also, it is not stated which period the anomalies are relative to. Even if it is mentioned in the main text, please repeat it in the figure caption.

A: Thank you. These changes have been made.

Figure 7: a), b), etc. missing on figure panels. Also, the colour legend is opposite in the top left and right panels, which is confusing when comparing the two. Moreover, the unit in the lower left panel is stated to be 'm', but dSSH/dx should be dimensionless.

A: Thank you for pointing this out. We have fixed the units (now using SSH gradient), and they are now m/km, or effectively unitless. Figure panels are also added.

Figure 8: The unit on the x-axis in the left panel (m/km) must be wrong. In the associated bottom left panel in Figure 7, the unit is 10-6 m. Please check.

A: Thank you, this has been fixed. The units of m/km is correct, and is also used now in Figure 7 as well.

Figure 10: Your analysis is based on model results, but here you state that this is a diagram of the observed ocean feedback loop. It would be more consistent to state that the diagram is showing the modelled ocean feedback loop.

A: Thank you, we have changed it to say the 'proposed ocean feedback loop'

Figure S3 label: The figure label states "decadally averaged Turbulent (top), Radiative (middle) and total (bottom) …", but it does not tell turbulent or total of what. I presume it is heat fluxes, but please state it explicitly.

A: Thank you, this has been fixed.